# The prevalence of mild-to-moderate distress in patients with end-stage renal disease: results from a patient survey using the emotion thermometers in four hospital Trusts in the West Midlands, UK

Sarah Damery,[1] Celia Brown,[2] Kim Sein,[1] Johann Nicholas,[3] Jyoti Baharani,[4] Gill Combes[1]

[1]Institute of Applied Health Research, University of Birmingham, Edgbaston, UK
[2]Warwick Medical School, University of Warwick, Coventry, UK
[3]Renal Unit, Shrewsbury and Telford NHS Trust, Shrewsbury, UK
[4]Renal Unit, Birmingham Heartlands Hospital, Heart of England NHS Foundation Trust, Birmingham, UK

**Correspondence to**
Dr Sarah Damery;
s.l.damery@bham.ac.uk

## ABSTRACT

**Objectives** To assess the prevalence of mild-to-moderate distress in patients with end-stage renal disease (ESRD) and determine the association between distress and patient characteristics.

**Design** Cross-sectional survey using emotion thermometer and distress thermometer problem list.

**Setting** Renal units in four hospital Trusts in the West Midlands, UK.

**Participants** Adult patients with stage 5 chronic kidney disease who were: (1) On prerenal replacement therapy. (2) On dialysis for less than 2 years. (3) On dialysis for 2 years or more (4) With a functioning transplant.

**Outcomes** The prevalence of mild-to-moderate distress, and the incidence of distress thermometer problems and patient support needs.

**Results** In total, 1040/3730 surveys were returned (27.9%). A third of survey respondents met the criteria for mild-to-moderate distress (n=346; 33.3%). Prevalence was highest in patients on dialysis for 2 years or more (n=109/300; 36.3%) and lowest in transplant patients (n=118/404; 29.2%). Prevalence was significantly higher in younger versus older patients ($\chi^2$=14.33; p=0.0008), in women versus men ($\chi^2$=6.63; p=0.01) and in black and minority ethnic patients versus patients of white ethnicity ($\chi^2$=10.36; p=0.013). Over 40% of patients (n=141) reported needing support. More than 95% of patients reported physical problems and 91.9% reported at least one emotional problem.

**Conclusions** Mild-to-moderate distress is common in patients with ESRD, and there may be substantial unmet support needs. Regular screening could help identify patients whose distress may otherwise remain undetected. Further research into differences in distress prevalence over time and at specific transitional points across the renal disease pathway is needed, as is work to determine how best to support patients requiring help.

## Strengths and limitations of this study

► This study is the first to explore the prevalence of mild-to-moderate distress in a large cohort of patients with end-stage renal disease (ESRD).

► Findings suggest that distress affects around a third of patients with ESRD at any one time, and that there are substantial unmet support needs in this population.

► The inclusion of multiple study sites is likely to have minimised any bias arising from variations in the organisation and delivery of renal services.

► The survey response rate was low, and younger patients, those from black and minority ethnic groups and patients who had been more recently diagnosed were under-represented in responses.

renal replacement therapy (RRT) in England.[1] Treatment is life-sustaining but not curative, and patients must constantly adjust to frequent changes to health status and likelihood of survival. Consequently, patients can experience many emotional and psychological stressors, including acceptance of diagnosis, disease progression, choosing treatment options, coping with dialysis and associated impacts on employment, relationships and lifestyle.[2 3] Evidence suggests that patients with ESRD experience rates of depression and anxiety markedly higher than the general population.[4 5] Establishing prevalence is challenging, partly because many uraemic symptoms can be misinterpreted as symptoms of depression,[6] and partly due to the variation in prevalence estimates obtained using different diagnostic tools and modes of assessment (eg, self-reported vs interview-based scales).[7] Consequently, depression

## BACKGROUND

In 2016, there were 51 672 patients with end-stage renal disease (ESRD) receiving

and anxiety prevalence estimates reported in the ESRD literature range from around 6% to 71%,[8] but are generally considered to be between 20% and 30% for dialysis patients[9] and around 25% for transplant patients.[10] This compares to a point prevalence of depression of between 2% and 9% and lifetime depression risk of around 7% in the general population.[11]

Untreated anxiety and depression in patients with ESRD are associated with decreased health-related quality of life and higher symptom burden.[12] These factors may raise the risk of poor outcomes, increased healthcare use and suboptimal adherence with diet and medication regimes.[13–15] There is also evidence that depression status is associated with an elevated risk of all-cause mortality in renal patients, with meta-analysis suggesting an excess mortality risk attributable to depression higher than that observed in other chronic diseases such as cancer and diabetes.[7] Provision of emotional and psychological support is central to the recommended management of renal disease, and both the UK Department of Health and National Institute for Health and Care Excellence mandate provision of such support within national renal guidelines.[16 17]

While supportive services are relatively advanced for those with higher-level needs requiring psychiatric or psychological intervention, there is a lack of robust data on the prevalence of lower-level support needs—defined as difficulties adjusting to, and coping effectively with, renal failure, diagnosis, physical symptoms and treatment. It may also be useful to broaden the focus beyond narrowly defined anxiety and depression to consider the determinants and consequences of more general emotional and psychological difficulty encompassed by the term 'distress'. While linear progression from lower-level to severe difficulty is not inevitable, timely identification of patient distress may facilitate effective management. This study aimed to assess the prevalence of mild-to-moderate distress in a cohort of patients with ESRD and determine the association between distress and sociodemographic and clinical characteristics. It forms part of a larger mixed-methods study that aims to understand how the recognition and management of patients' emotional and psychological support needs can be integrated into the ESRD pathway.[18]

## Methods

The study used a cross-sectional survey, distributed by post to all eligible patients with ESRD managed at four National Health Service hospital Trusts in the West Midlands, UK. Trusts were chosen to maximise diversity in patient demographics, catchment size, urban-rural mix and the organisation of psychological support services: site 1 (small) and site 2 (large) both serve urban, inner city catchments with substantial black and minority ethnic (BME) populations. Site 3 (medium) and site 4 (large) both serve predominantly white populations in urban areas with surrounding rural districts. Sites 2 and 4 have

access to a renal psychologist for the provision of psychological support services, whereas sites 1 and 3 do not.

## Participants and recruitment

Eligible patients were aged 18+ years, diagnosed with stage 5 chronic kidney disease (CKD), and grouped according to their stage on the ESRD pathway: (1) Pre-RRT. (2) On peritoneal or haemodialysis for less than 2 years. (3) On dialysis for 2 years or more. (4) With a functioning transplant. Although differentiating according to dialysis vintage is not a recognised clinical distinction, it was hypothesised by the clinicians involved in designing this study that there may be differences in distress prevalence between patients initiating dialysis more recently versus those undergoing dialysis for longer. Patients using psychiatric services since stage 5 CKD diagnosis were excluded. Renal unit staff at each Trust identified eligible patients from hospital records, and survey packs were prepared on Trust premises by the university research team. Eligible patients received a letter of invitation from the renal unit's lead consultant, an information sheet and a survey, to be returned directly to the researchers. Return envelopes were marked with a unique identifier for recording returns, and non-responders received one reminder after 6 weeks. Mailings were carried out between January 2016 and May 2017.

## Survey

The survey measured distress with emotion thermometers (ETs),[19 20] which use a Visual Analogue Scale covering five domains: distress, anxiety, depression, anger and perceived need for help. Patients score each domain on an 11-point Likert Scale to rate their levels of emotional upset during the preceding week, where '0' corresponds to none and '10' denotes extreme problems. Although not validated specifically for use with renal patients, the ET has been widely used in studies of patients with cancer and other chronic conditions where it has been found to be sensitive in identifying emotional difficulty and broadly defined distress.[19] It incorporates the distress thermometer (DT), which has been validated in the UK population with CKD.[21] The survey also included the DT problem list[22] which lists 36 problems across five domains: practical (n=5), family (n=3), emotional (n=6), spiritual (n=1) and physical (n=21). Patients indicated which (if any) of the 36 problems they had experienced in the previous week. Closed questions covered sociodemographic characteristics (age, sex, ethnicity) and treatment modality (where relevant).

## Data analysis

Thresholds for distress using the ET have been validated,[19 20] with a score of 4–5 denoting mild distress, and 6–7 denoting moderate distress. Patients were defined as having mild-to-moderate distress if they: (1) Scored between 4 and 7 on the DT (regardless of scores in the other ET domains), or (2) scored 0–3 on the DT and 4–7 on one or more of the anxiety, depression and anger

thermometers, with no thermometer exceeding 7. Analysis was descriptive, focusing on associations between distress and respondents' stage on the ESRD pathway and sociodemographic characteristics. Anonymised data were obtained from hospital information systems to allow a comparison of the characteristics of respondents and non-respondents on the basis of age group, sex, ethnic group and ESRD pathway stage. Comparisons were undertaken using $\chi^2$ analysis. Any surveys in which the ETs were left blank by a respondent were excluded from analysis. The prevalence of total and individual problems from the DT problem list were analysed descriptively, and medians and IQRs were calculated to compare numbers of problems cited within each domain. Data were analysed using SPSS V.21.0 (IBM Corp, Armonk, New York, USA).

### Sample size

The primary outcome was the difference in the prevalence of mild-to-moderate distress across patients at different ESRD stages. We anticipated an average prevalence of 25% across all patients,[9] with patients in the stages with the highest and lowest prevalence at ±5% points from this average (ie, 20% for the stage with the lowest prevalence and 30% for the stage with the highest prevalence). This equated to a small effect size (w) of approximately 0.1. To detect this difference with 80% power and 5% significance, a total of 1090 responses were required (assuming approximately equal numbers of patients in each ESRD stage).

### Patient and public involvement

The study design and research questions were developed with inputs from a study advisory group that included patient representatives, and the patient and public involvement group attached to the chronic diseases theme of Collaboration for Leadership in Applied Health Research and Care (CLAHRC) West Midlands. These groups were also involved in selecting appropriate outcome measures that optimised data quality while minimising participant burden. Patients were not involved in recruitment to the study or its conduct. Study participants will be sent a plain English summary of final study results if they indicated during the informed consent process that they would like to receive this.

## RESULTS

A total of 3730 surveys were sent across the four study sites. One hundred patients died between the initial and reminder mailings (2.7%) and 2442 recipients (65.5%) did not respond. Of the 1188 responses received (31.8%), 148 were removed due to non-completion of ETs (4.0%), giving a total of 1040 valid responses (27.9%) (figure 1). Rates of valid responses ranged from 23.0% in site 1% to 30.4% in site 4. Younger patients (<65 years old) were significantly less likely to respond than those aged 65 years and over, as were those from BME groups compared with those in the white ethnic group. Patients yet to begin RRT and those on dialysis for less than 2 years were significantly less likely to respond than patients with a transplant or who had been undergoing dialysis for 2 years or more. There was no significant difference between responders and non-responders on the basis of sex (table 1).

### Characteristics of respondents

The majority of respondents were male (n=633; 60.9%) and in the white ethnic group (n=902; 86.7%) (table 2). Patients aged between 51 years and 69 years constituted the largest age group (n=441; 42.9%), with those aged under 50 years comprising 16.9% of the total (n=174). Nearly two-fifths of respondents had received a transplant (n=404; 38.8%) and 28.8% had been on dialysis for 2 years or more (n=300). Of the 454 patients undergoing

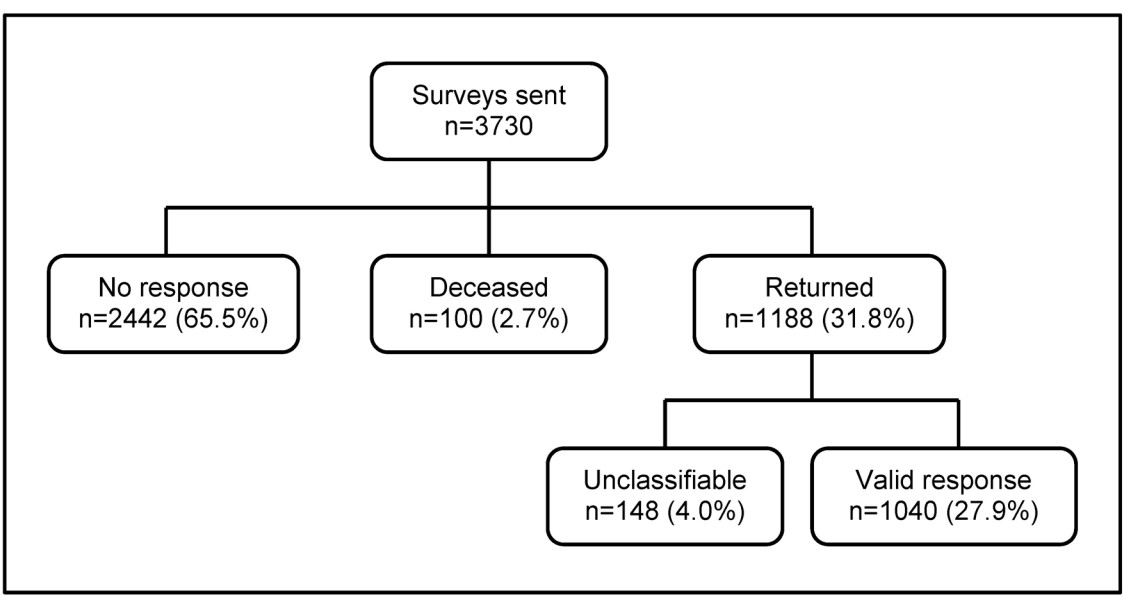

**Figure 1** Surveys mailed and returned.

**Table 1** Comparison of the characteristics of survey responders and non-responders

| Characteristic | Respondents (%) | Non-respondents (%) | Total | Comparison of proportions |
|---|---|---|---|---|
| **Stage on ESRD pathway** | | | | |
| Pre-RRT | 183 (24.6) | 560 (75.4) | 743 | **$\chi^2$=9.96; p=0.02** |
| Dialysis <2 years | 162 (25.2) | 481 (74.8) | 643 | |
| Dialysis 2+ years | 293 (30.0) | 684 (70.0) | 977 | |
| Transplant | 402 (29.4) | 965 (70.6) | 1367 | |
| **Age group*** | | | | |
| Less than 65 years | 503 (24.7) | 1537 (75.3) | 2040 | **$\chi^2$=19.86; p<0.0001** |
| 65 years and above | 524 (31.3) | 1151 (68.7) | 1675 | |
| **Sex** | | | | |
| Male | 635 (28.0) | 1636 (72.0) | 2271 | $\chi^2$=0.01; p=0.92 |
| Female | 405 (27.8) | 1054 (72.2) | 1459 | |
| **Ethnicity†** | | | | |
| White | 902 (34.4) | 1720 (65.6) | 2622 | **$\chi^2$=76.16; p<0.0001** |
| Black and minority ethnic | 138 (17.9) | 635 (82.1) | 773 | |

*Age unknown for 15 patients;
†Ethnicity unknown for 335 patients;
Bold text indicates statistical significance.
ESRD, end-stage renal disease; RRT, renal replacement therapy.

regular dialysis, the most common modality was hospital/incentre haemodialysis (n=343; 75.6%).

### Prevalence of distress

A total of 561 respondents (53.9%) were categorised as having none-to-low distress, and a further 133 patients (12.3%) met the criteria for severe distress (ie, scoring >7 on one or more of the ETs). A total of 346 respondents (33.3%) met the criteria for mild-to-moderate distress (hereafter termed 'distress'). Distress prevalence by hospital site was lowest in site 4 (30.9%) and highest in site 1 (38.5%), although the difference between study sites was not statistically significant ($\chi^2$=3.24; p=0.356). Distress was identified in 208 patients (60.1%) on the basis of their DT score alone, with the remaining 138 (39.9%) identified from their scores on the anxiety, depression or anger thermometers. Distress prevalence was highest in patients who had been on dialysis for 2 years or more (n=109/300; 36.3%) and lowest in transplant patients (n=118/404; 29.2%). For dialysis modality, distress was most prevalent for home haemodialysis patients (n=13/31; 41.9%) and lowest in the peritoneal dialysis group (n=22/80; 27.5%), although numbers were small. There was no significant difference in rates of distress on the basis of ESRD stage or dialysis type when groups were compared. In contrast, all sociodemographic characteristics showed significant differences in distress prevalence between groups. Patients in the youngest age group were significantly more likely to report distress than those in the older age groups (n=78/174; 44.8%; $\chi^2$=14.33; p=0.0008). The prevalence of distress was significantly higher for women than

men (38.1% vs 30.2%; $\chi^2$=6.63; p=0.01) and for BME patients compared with white patients (45.7% vs 31.4%; $\chi^2$=10.36; p=0.0013).

### Perceived need for help

Scores on the 'need' thermometer were assessed as an outcome measure indicating patients' perceived need for help. Scores of 4 and above were considered to indicate unmet support needs[19] (table 3). Patients at study sites 1 and 3 had significantly higher perceived support needs than those in sites 2 and 4 (29.4% and 22.9% vs 21.5% and 20.1%; $\chi^2$=9.49; p=0.02). There were also significant differences by subgroup according to ESRD stage: 66.0% of patients on dialysis for less than 2 years (n=35) required support, as did 55.7% of patients on dialysis for 2 years or more (n=59). Perceived support needs in the pre-RRT and transplant groups were significantly lower, at 34.9% and 29.9% respectively ($\chi^2$=27.71; p<0.0001). Support needs were also significantly higher in BME compared with white patients, at 57.6% vs 37.9% ($\chi^2$=7.06; p=0.008).

### DT problem list

All but six patients reported experiencing at least one problem on the DT problem list in the preceding week (98.0%) (table 4). The most frequently reported problems were all in the emotional and physical domains, with 91.9% of patients reporting at least one emotional problem in the previous week, and 95.1% reporting at least one physical problem. 'Worry' was the the most common problem across all domains (n=247; 74.3%). 'Fatigue' (71.7%), 'dry/itchy skin' (65.0%) and 'sleep'

**Table 2** Respondent characteristics and proportion by sub-group with mild-to-moderate distress

| Characteristic | Respondents (%) | Mild-to-moderate distress (%) | Comparison of proportions (mild-to-moderate distress patients)* |
|---|---|---|---|
| All respondents | 1040 (100.0) | 346 (33.3) | |
| Study site | | | |
| Site 1 | 187 (18.0) | 72 (38.5) | $\chi^2$=3.24; p=0.356 |
| Site 2 | 177 (17.0) | 58 (32.8) | |
| Site 3 | 323 (31.1) | 107 (33.1) | |
| Site 4 | 353 (33.9) | 109 (30.9) | |
| Stage on ESRD pathway | | | |
| Pre-RRT | 182 (17.5) | 64 (35.2) | $\chi^2$=4.85; p=0.183 |
| Dialysis <2 years | 154 (14.8) | 55 (35.7) | |
| Dialysis 2+ years | 300 (28.8) | 109 (36.3) | |
| Transplant | 404 (38.8) | 118 (29.2) | |
| *Dialysis type* (n=454) | | | |
| Hospital/incentre haemodialysis | 343 (75.6) | 129 (37.6) | $\chi^2$=3.36; p=0.186 |
| Home haemodialysis | 31 (6.8) | 13 (41.9) | |
| Peritoneal dialysis | 80 (17.6) | 22 (27.5) | |
| Age group† | | | |
| Less than 50 years | 174 (16.9) | 78 (44.8) | **$\chi^2$=14.33; p=0.0008** |
| 50–69 years | 441 (42.9) | 145 (32.9) | |
| 70 years and above | 414 (40.2) | 119 (28.7) | |
| Sex | | | |
| Male | 633 (60.9) | 191 (30.2) | **$\chi^2$=6.63; p=0.01** |
| Female | 407 (39.1) | 155 (38.1) | |
| Ethnicity | | | |
| White | 902 (86.7) | 283 (31.4) | **$\chi^2$=10.36; p=0.0013** |
| Black and minority ethnic | 138 (13.3) | 63 (45.7) | |

*Bold text denotes a statistically significant difference between groups;
†Age unknown for four respondents
ESRD, end-stage renal disease; RRT, renal replacement therapy.

(61.3%) were the most common physical problems. 'Spiritual problems' (6.6%) and 'problems with childcare' (family domain: 3.8%) were reported least frequently. The median number of total problems reported was 12/36 (IQR: 7 to 16) (table 5).

Subgroup analysis shows that transplant patients reported significantly fewer physical problems and total problems than pre-RRT or dialysis patients (median physical problems 6/21, IQR 3 to 10, p=0.019; median total problems 10/36, IQR 5 to 16, p=0.023). Patients 70 years and older reported significantly fewer problems on the practical, family and emotional domains than younger patients. They also reported significantly fewer total problems. Women reported significantly more physical problems than men (median 8/21, IQR 5 to 12 vs 7/21, IQR 5 to 11; p=0.043).

## DISCUSSION

Detecting distress is important for the optimal care of patients with ESRD, and evidence suggests that reducing emotional and psychological difficulties may enhance well-being and improve patients' ability to engage with complex and demanding treatments. Our study has shown that as many as one in three patients with ESRD may have mild-to-moderate distress, and rates of reporting emotional and physical problems were high. This was evident across the ESRD pathway, with around 35% of pre-RRT and dialysis patients and 29% of transplant patients meeting the criteria for mild-to-moderate distress. The finding that the prevalence of distress was almost as high in transplant patients as those undergoing dialysis suggests that although they may experience fewer physical problems, the need for ongoing psychological

**Table 3** Perceived need for help with distress by subgroup with mild-to-moderate distress (n=346)

| Characteristic | Median 'need' score (IQR) | Patients scoring 4 or more on need thermometer (%) | Comparison of proportions (mild-to-moderate distress patients)* |
|---|---|---|---|
| All respondents | 3 (1 to 5) | 141 (40.8) | |
| **Study site** | | | |
| Site 1 | 1 (0 to 4) | 55 (29.4) | **$\chi^2$=9.49; p=0.02** |
| Site 2 | 1 (0 to 3) | 38 (21.5) | |
| Site 3 | 1 (0 to 3) | 74 (22.9) | |
| Site 4 | 0 (0 to 3) | 71 (20.1) | |
| **Stage on ESRD pathway** | | | |
| Pre-RRT | 2 (0 to 4) | 22 (34.9) | **$\chi^2$=27.71; p<0.0001** |
| Dialysis <2 years | 3 (2 to 6) | 35 (66.0) | |
| Dialysis 2+ years | 4 (2 to 6) | 59 (55.7) | |
| Transplant | 2 (0 to 4) | 35 (29.9) | |
| **Dialysis type (n=454)** | | | |
| Hospital/in-centre haemodialysis | 4 (2 to 6) | 70 (56.0) | $\chi^2$=5.44; p=0.07 |
| Home haemodialysis | 4 (2 to 8) | 7 (53.8) | |
| Peritoneal dialysis | 3 (0 to 5) | 6 (28.6) | |
| **Age group** | | | |
| Less than 50 years | 3 (2 to 5) | 35 (44.9) | $\chi^2$=0.85; p=0.654 |
| 50–69 years | 3 (1 to 5) | 55 (39.0) | |
| 70 years and above | 3 (1 to 5) | 48 (41.4) | |
| **Sex** | | | |
| Male | 3 (1 to 5) | 73 (39.2) | $\chi^2$=0.54; p=0.462 |
| Female | 3 (1 to 5) | 67 (43.8) | |
| **Ethnicity** | | | |
| White | 3 (1 to 5) | 106 (37.9) | **$\chi^2$=7.06; p=0.008** |
| Black and minority ethnic | 5 (2 to 7) | 34 (57.6) | |

*Bold text indicates a statistically significant difference between groups.
ESRD, end-stage renal disease; RRT, renal replacement therapy.

adjustment does not end at the point of transplantation.[10] Mild-to-moderate distress was most prevalent in the group of patients who had been on dialysis for 2 years or more (36.3%), which may reflect a lack of adjustment and coping in this group with the ongoing challenges of undergoing regular dialysis treatment over an extended period, potentially declining health, and the limitations that dialysis treatment places on family, relationships and lifestyle. There was also variation by dialysis type, with patients receiving hospital/incentre haemodialysis reporting distress rates of 37.6%, compared with 27.5% for patients undergoing peritoneal dialysis. Numbers were too small to detect statistically significant differences, but this distinction has been found elsewhere,[23] and would benefit from further study.

Distress prevalence was strongly associated with sociodemographic characteristics and was significantly higher in younger versus older patients, in BME versus white patients and in women compared with men. These trends have also been found in studies of anxiety and depression in renal patients.[24 25] There are numerous psychological theories of health and illness which may have a role in explaining patients' variable responses to ESRD diagnosis and treatment.[26–28] These theories emphasise that experience of distress is likely to be determined by an individual's personal degree of resilience and individual coping resources rather than being associated with specific clinical characteristics.

Comparable national figures for the pre-RRT group are not available, but if the prevalence of mild-to-moderate distress found in this study was standardised to the current population in England with a functioning transplant or undergoing dialysis, it would equate to 18 970 patients with ESRD experiencing difficulties, of whom 7835 may want help.[1] It has been argued that the primary goal of supportive services should be to

**Table 4** Number and proportion of patients reporting problems on the distress thermometer problem list

| Problems by domain | Number of patients (%)* | Patients reporting at least one problem for the domain (%) |
|---|---|---|
| Any problems (n=36) | – | 339 (98.0) |
| Practical domain (n=5) | | |
| Transport | 84 (24.3) | |
| Insurance/financial | 76 (22.0) | |
| Work | 53 (15.3) | 167 (48.3) |
| Housing | 30 (8.7) | |
| Childcare | 13 (3.8) | |
| Family domain (n=3) | | |
| Dealing with friend/relative | 66 (19.1) | |
| Dealing with partner | 60 (17.3) | 118 (34.1) |
| Dealing with children | 27 (7.8) | |
| Emotional domain (n=6) | | |
| Worry | 257 (74.3) | |
| Loss of interest in usual activities | 186 (53.8) | |
| Sadness | 166 (48.0) | 318 (91.9) |
| Depression | 159 (46.0) | |
| Nervousness | 150 (43.4) | |
| Fears | 148 (42.8) | |
| Spiritual domain (n=1) | | |
| Spiritual/religious concerns | 23 (6.6) | – |
| *Physical domain (n=21)* | | |
| Fatigue | 248 (71.7) | |
| Skin dry/itchy | 225 (65.0) | |
| Sleep | 212 (61.3) | |
| Memory/concentration | 183 (52.9) | |
| Pain | 178 (51.4) | |
| Getting around | 171 (49.4) | |
| Tingling in hands/feet | 144 (41.6) | |
| Breathing | 133 (38.4) | |
| Feeling swollen | 127 (36.7) | |
| Bathing/dressing | 119 (34.4) | 329 (95.1) |
| Appearance | 116 (33.5) | |
| Eating | 111 (32.1) | |
| Nose dry/congested | 108 (31.2) | |
| Constipation | 106 (30.6) | |
| Nausea | 100 (28.9) | |
| Changes in urination | 94 (27.2) | |
| Indigestion | 92 (26.6) | |
| Sexual | 88 (25.4) | |
| Diarrhoea | 79 (22.8) | |
| Mouth sores | 42 (12.1) | |
| Fevers | 31 (9.0) | |

*Problems within each domain ranked from most to least frequently cited by respondents

**Table 5** Median number of problems reported in each distress thermometer problem list domain, by patient group

| Characteristic | Practical*,† | Family | Emotional | Physical | Total |
|---|---|---|---|---|---|
| | Median (IQR) | Median (IQR) | Median (IQR) | Median (IQR) | Median (IQR) |
| **Stage on ESRD pathway** | | | | | |
| Pre-RRT | **1 (0 to 1)** | **0 (0 to 1)** | 3 (1 to 4) | **8 (4 to 12)** | **12 (7 to 17)** |
| Dialysis <2 years | **1 (0 to 2)** | **0 (0 to 1)** | 3 (2 to 5) | **8 (6 to 12)** | **13 (9 to 17)** |
| Dialysis 2+ years | **1 (0 to 1)** | **0 (0 to 1)** | 3 (2 to 5) | **9 (6 to 11)** | **13 (9 to 16)** |
| Transplant | **0 (0 to 1)** | **0 (0 to 1)** | 3 (1 to 4) | **6 (3 to 10)** | **10 (5 to 16)** |
| | **P<0.0001** | **P=0.001** | **P=0.038** | **p=0.019** | **p=0.023** |
| **Dialysis type (n=164)** | | | | | |
| Hospital/incentre haemodialysis | 1 (0 to 1) | 0 (0 to 1) | 3 (2 to 5) | 8 (5 to 11) | 13 (8 to 16) |
| Home haemodialysis | 1 (0 to 2) | 0 (0 to 1) | 4 (2 to 5) | 10 (8 to 13) | 16 (11 to 19) |
| Peritoneal dialysis | 0 (0 to 1) | 0 (0 to 1) | 3 (2 to 4) | 8 (6 to 11) | 12 (9 to 16) |
| *Age group*[4] | | | | | |
| Less than 50 years | **1 (0 to 2)** | **0 (0 to 1)** | **3 (2 to 5)** | 7 (4 to 11) | **13 (8 to 17)** |
| 50–69 years | **1 (0 to 2)** | **0 (0 to 1)** | **3 (2 to 5)** | 8 (5 to 11) | **12 (8 to 18)** |
| 70 years and above | **0 (0 to 1)** | **0 (0 to 0)** | **3 (1 to 4)** | 7 (4 to 10) | **11 (7 to 15)** |
| | | **P=0.033** | **P=0.038** | | **P=0.033** |
| **Sex** | | | | | |
| Male | 1 (0 to 1) | 0 (0 to 1) | 3 (2 to 4) | **7 (4 to 10)** | 12 (7 to 16) |
| Female | 0 (0 to 1) | 0 (0 to 1) | 3 (2 to 5) | **8 (5 to 12)** | 12 (8 to 17) |
| | | | | **P=0.043** | |
| **Ethnicity** | | | | | |
| White | 0 (0 to 1) | 0 (0 to 1) | 3 (2 to 4) | 8 (5 to 11) | 12 (7 to 16) |
| Black and minority ethnic | 0 (0 to 2) | 0 (0 to 1) | 4 (1 to 5) | 7 (4 to 10) | 11 (7 to 17) |
| All respondents (n=346) | 0 (0 to 1) | 0 (0 to 1) | 3 (2 to 5) | 8 (5 to 11) | 12 (7 to 16) |

*Bold text denotes a statistically significant difference between groups (Mann–Whitney test for characteristics with two subgroups; Kruskall–Wallis test for characteristics with three or more subgroups);
†Spiritual domain excluded from analysis.
ESRD, end-stage renal disease; RRT, renal replacement therapy.

distinguish between transient and ongoing difficulties and focus on optimising patients' ability to cope with their condition.[29] In this context, it is likely that the most feasible supportive interventions are those that can be incorporated into routine clinical practice and standard protocols for ESRD care. ETs could be used regularly by clinicians during patient consultations or by renal nurses to screen for distress. Screening in itself will not necessarily lead to better patient outcomes, but it may help to identify patients whose distress would otherwise remain undetected, and in doing so, could enable the targeted provision of support services.

Experiencing distress may be considered a normal response to chronic disease diagnosis, and not all patients with mild-to-moderate distress will want to receive support. The likelihood that patients with ESRD will experience symptoms of distress was predominantly associated with age, sex and ethnicity. Consequently, screening for distress would seem particularly important for younger patients, who may experience significantly greater disruption to their family and working life than older patients, women, and for patients in BME groups, who reported uniformly high rates of mild-to-moderate distress and high support needs. It may also be useful to target screening for distress towards those who undergo regular hospital or incentre haemodialysis. Providing appropriate information to patients early on in the ESRD pathway (pre-RRT) about the distress they may experience could also help to manage patients' expectations and allow timely identification of emotional and psychological difficulties. There was also evidence of a centre effect in the proportion of patients reporting a perceived need for support at participating study sites. Patients in sites 1 and 3, where there is no access to a renal psychologist, reported significantly higher rates of support need than those in sites 2 and 4 where renal psychologist support is available. This suggests that the availability of inhouse psychological support services may play an important role in helping patients with ESRD to manage distress.

### Limitations

The survey response rate was low, and younger patients, those from BME groups and patients who had been more recently diagnosed were under-represented in responses. The study was marginally underpowered, with 1040 valid responses received, rather than the 1090 required by our a priori sample size estimation. We are also unable to say whether distress is more or less common in survey respondents compared with non-respondents, nor could we perform detailed subgroup analyses for patients within each ESRD pathway stage. This meant that participants in each ESRD pathway stage were treated as a homogeneous cohort when in reality there may have been differences between them that may have impacted on their experience of distress such as issues with medication or ESRD-related complications like fistula failure, infection or transplant failure. However, in-depth qualitative research was undertaken with renal patients to explore the detailed determinants and consequences of their distress in a linked component of this study (findings to be reported elsewhere).

Nevertheless, our findings with regard to the patient groups most likely to experience distress were similar to existing evidence on anxiety and depression in renal disease. Because younger, BME patients were under-represented in our sample, the finding that patients in these groups reported significantly higher rates of mild-to-moderate distress than older, white patients, suggests that we may have underestimated rather than overestimated overall distress prevalence. The inclusion of multiple study sites is likely to have minimised any bias that may arise from variations in the organisation and delivery of renal services. Patients referred to psychiatric services (as noted in their hospital record) were excluded, but we cannot know whether our sample included patients who had independently sought counselling or support via their general practitioner. The study was also cross-sectional rather than longitudinal, and consequently allows limited understanding of the relationship between time since ESRD diagnosis and ability to cope with the resulting stressors. Some studies with renal patients have found no correlation between depressive symptoms and time since treatment initiation,[9] whereas others have found a tendency for depression status to worsen.[30] Future work using a cohort study design would aid understanding of the ways that individuals adapt to ESRD diagnosis and its ongoing management over time.

### CONCLUSION

This study is, to our knowledge, the first to explore the prevalence of mild-to-moderate distress in a large cohort of patients with ESRD. Our findings show that mild-to-moderate distress is common—even after transplantation—and there may be substantial unmet patient support needs. Further research into possible variations in distress prevalence over time and at different transitional points across the ESRD pathway is needed, as is work to determine how best to support patients requiring help.

**Acknowledgements** The authors thank the renal patient advisory group and the patient and public involvement group of the chronic disease theme of CLAHRCWM for their contribution to study design and choice of data collection tools.

**Contributors** SD managed the survey, undertook all statistical analyses and drafted the first version of the manuscript. CB contributed to research design and helped develop the methodological/analytical approach. KS contributed to data analysis, interpretation and refinement of study methodology. JN and JB both participated in research design and in the development and refinement of study methodology. GC conceived the study, was chief investigator, and contributed to all aspects of research design and methodology. All authors have seen and approved the final version of the manuscript.

**Funding** This research was funded by the UK National Institute for Health Research (NIHR) Collaboration for Leadership in Applied Health Research and Care West Midlands (CLAHRCWM). The views expressed in this manuscript are those of the authors and not necessarily those of the National Health Service, the NIHR or the Department of Health and Social Care. The funder had no role in study design, the collection, analysis and interpretation of data, or in writing the manuscript.

**Competing interests** None declared.

**Ethics approval** Ethic s approval was obtained from the Coventry and Warwickshire Research Ethic s Committee in October 2015 (Ref 15/WM/0288). The study was also approved by the Research Governance office of each of the participating hospital Trusts.

**Provenance and peer review** Not commissioned; externally peer reviewed.

**Data sharing statement** No additional data are available.

**Author note** Celia Taylor at the time the research was undertaken.

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
