## [Reviewer comments · BMJ Open]

ARTICLE DETAILS

TITLE (PROVISIONAL)	The prevalence of mild to moderate distress in patients with end stage renal disease: results from a patient survey using the emotion thermometers in four hospital Trusts in the West Midlands, UK
AUTHORS	Damery, Sarah; Brown, Celia; Sein, Kim; Nicholas, Johann; Baharani, Jyoti; Combes, Gill

VERSION 1 - REVIEW

REVIEWER	Stephanie M. Toth-Manikowski University of Illinois at Chicago U.S.A.
REVIEW RETURNED	14-Dec-2018

GENERAL COMMENTS	The authors investigate a topic that is relevant to any nephrologist who has ever taken care of patients with severely advanced CKD. The topic of "distress" as it is termed here is ubiquitous in a dialysis unit and as such, is a topic that is highly relevant to the field of nephrology. The following recommendations are ones that would help make the manuscript more easily readable in certain parts and also to clarify minor points that the average reader might have. 1) Under the "Survey" subheading, it is initially confusing which validated tools were administered in the survey. It would be better to have a sentence that states more simply, "Our survey consisted of two validated tools, XXX and XXX..." and then go on to describe each of the surveys as you do.2) Has the ET been validated in CKD patients? If not, please mention this. In which population was it validated in? Why is it ok to apply to a CKD/ESRD/transplant population?3) Under "Distress thermometer problem list," 'problems with childcare' is mentioned. Is there any way to know what percentage of people surveyed have young children?4) The last sentence of the second paragraph in the Discussion is confusing. Please re-word.5) The second and third sentences of the third paragraph in the Discussion is also somewhat confusing. Please consider re-wording.6). The second sentence in the fourth paragraph of the Discussion is confusing, specifically the part that starts with "the
--

	complementary finding that...". What complementary finding is being referred to here? 7) Please put percentages in the Respondents and Non-respondents column in Table 1.
--	--

REVIEWER	Soham Rej McGill University, Canada None Declared, with respect to this manuscript - I receive investigator-initiated research grant from Satellite Healthcare
REVIEW RETURNED	10-Mar-2019

GENERAL COMMENTS	This is a well written report about an important and relatively poorly understood topic - distress in ESRD patients. The sample size is quite solid esp. considering the limited previous research in this field. The findings are impressive and consistent with previous reports (33.3% of patients have mild-to-moderate distress), and identified important potential correlates of distress, and highlighted pertinent limitations (e.g. certain ethnicities and younger patients not as likely to answer the survey) I gladly suggest it for potential publication in BMJ Open with some minor modifications:  - Intro is solid -Methods: Page 5, Survey Subsection - Great that the Emotion Thermometers ETs are a validated and a visual analogue scale, in theory, should be easier for patients. Did you have other scales too (e.g. HADS, PHQ9, GAD7)? - Methods: Page 5/6, Data Analysis - although the chi-squared analyses are quite informative and this is a cross-sectional sample, I'm wondering whether the authors would consider a multi-variable logistic regression to assess whether the factors identified are independently correlated with distress. This is completely optional of course. - Limitations: Although for a mail-in survey in this patient population, a 31.8% response rate is quite reasonable - the response rate is an important limitation - it is hard to know about the majority of participants who didn't respond - were they more/less distressed - this probably had effects on the estimated prevalence of distress and correlates.
--

VERSION 1 – AUTHOR RESPONSE

Reviewer 1:

1. Under the "survey" subheading, it is initially confusing which validated tools were administered in the survey. It would be better to have a sentence that states more simply. "Our survey consisted of two validated tools, xxx and xxx..." and then go onto describe each of the surveys as you do.

Response: Apologies for confusion here – we have made changes to the "Survey" paragraph on page 5 as suggested to clarify and simplify the description of the tools used within the survey.

2. Has the ET been validated in CKD patients? If not, please mention this. In which population was it validated in? Why is it OK to apply to a CKD/ESRD/transplant population?

Response: Very few emotional measurement tools have been validated for use with renal patients. The ET has not been validated for CKD specifically, but has been used widely with cancer patients and patients with other chronic conditions. A key component of the ET is the distress thermometer, which has been validated in the CKD population in the UK (see new reference 21). As it was important for our study that we were able to measure broad emotional and psychological difficulty rather than specific elements of anxiety and depression that feature in other validated measures like the HADS or BDI, we chose to use the ET given that this was an easily comprehended visual tool, appropriate for incorporation into a patient survey, and included the DT which we knew had been fully validated.

The fact that the DT has been validated in the UK renal population and that the ET incorporates the DT has been clarified in the "Survey" paragraph on page 5.

3. Under "Distress thermometer problem list" 'problems with childcare' is mentioned. Is there any way to know what percentage of people surveyed have young children?

Response: Unfortunately, we did not ask patients about their personal circumstances/characteristics beyond collecting data on age, gender, ethnic group etc. So, we do not know what percentage of people had young children. No changes have been made to the manuscript in the light of this comment.

4. The last sentence of the second paragraph in the Discussion is confusing. Please re-word.

Response: Apologies for confusion, the sentence in question has been deleted, as it does not add anything further to what has already been said in the paragraph.

5. The second and third sentences of the third paragraph in the Discussion is also somewhat confusing. Please consider re-wording.

Response: Apologies. The second sentence has been deleted, as we agree that this is confusing and superfluous to the argument being made. The first part of the third sentence has been moved to the first sentence of the following paragraph where it fits better. This shortens the third sentence referred to above, which we feel makes it easier for the reader to understand.

6. The second sentence in the fourth paragraph of the Discussion is confusing, specifically the part that starts with "the complementary finding that..." What complementary finding is being referred to here?

Response: We agree that this sentence is confusing. It has been split in two, with the first half addressing issues of age, gender and ethnicity. The second half addresses issues pertinent to dialysis patients. We feel that this reorganisation of the text makes the sentence in question (and the paragraph as a whole) far clearer.

7. Please put percentages in the Respondents and Non-respondents column in Table 1.

Response: Percentages have been added to Table 1 as requested.

Reviewer 2:

1. Methods: Page 5, Survey subsection – Great that the Emotion Thermometers are a validated and a visual analogue scale, in theory, should be easier for patients. Did you have other scales too (e.g. HADS, PHQ9, GAD7).

Response: We did not use any other scales in our survey, as our intention was to use a global measure of distress rather than other measures which were more sensitive towards identifying anxiety and depression, and where there were less clear thresholds for identifying patients who had lower level difficulties (i.e. mild to moderate rather than severe distress). No changes to the manuscript have been made as a result of this comment, although it is hoped that this issue has been clarified as part of our response to Reviewer 1's comments about the "Survey" subsection (reviewer 1 comments 1 and 2)

2. Methods: Page 5/6, Data Analysis – although the chi squared analyses are quite informative and this is a cross-sectional sample, I'm wondering whether the authors would consider a multi-variable logistic regression to assess whether the factors identified are independently correlated with distress. This is completely optional of course.

Response: Thank you for this suggestion, and it was something we considered when preparing this manuscript. However, we categorised survey respondents into three groups (no distress, mild-moderate distress and severe distress), thus distress was not a binary outcome unless the no distress and severe distress groups were combined, which we did not feel was appropriate. We could have undertaken a multinomial logistic regression to see which variables were associated with being in the mild-moderate group compared to either of the other groups, but we did not feel that this would give us very different information than that already provided with the existing chi-squared analysis. Therefore, we have chosen not to pursue this option.

3. Limitations: Although for a mail-in survey in this patient population, a 31.8% response rate is quite reasonable – the response rate is an important limitation – it is hard to know about the majority of participants who didn't respond – where they more or less distressed – this probably had effects on the estimated prevalence of distress and correlates.

Response: Thank you for this point, which we strongly agree with. We have explored the issue of the response rate and the fact that we cannot know whether there was a difference in rates of distress in responders vs. non-responders (first paragraph of limitations section, page 9), so we have not made any further changes to the manuscript following this point.